# The Potential Role of Polyamines in Epilepsy and Epilepsy-Related Pathophysiological Changes

**DOI:** 10.3390/biom12111596

**Published:** 2022-10-29

**Authors:** Jiayu Liu, Zhi Yu, Buajieerguli Maimaiti, Qian Meng, Hongmei Meng

**Affiliations:** Department of Neurology and Neuroscience Center, The First Hospital of Jilin University, NO. 1 Xinmin Street, Changchun 130021, China

**Keywords:** polyamine metabolism, epilepsy, seizure, pathological change, neuroprotection, neurotoxicity

## Abstract

Epilepsy is one of the most common neurological disorders and severely impacts the life quality of patients. Polyamines are ubiquitous, positively charged aliphatic amines that are present at a relatively high level and help regulate the maintenance of cell membrane excitability and neuronal physiological functions in the central nervous system. Studies have shown abnormalities in the synthesis and catabolism of polyamines in patients with epilepsy and in animal models of epilepsy. The polyamine system seems to involve in the pathophysiological processes of epilepsy via several mechanisms such as the regulation of ion permeability via interaction with ion channels, involvement in antioxidation as hydroperoxide scavengers, and the induction of cell damage via the production of toxic metabolites. In this review, we try to describe the possible associations between polyamines and epilepsy and speculate that the polyamine system is a potential target for the development of novel strategies for epilepsy treatment.

## 1. Introduction

Polyamines (PAs) are ubiquitous biogenic amines with low molecular weights, and classically refer to three molecules: putrescine (PUT), spermidine (SPD), and spermine (SPM). Polyamines are completely protonated under physiological conditions (Figure 1). As polycations, polyamines can interact with various negatively charged cellular macromolecules such as nucleic acids, ATP, proteins, and phospholipids, and thus play a critical role in multiple biological processes, including the stabilization of nucleic acids, gene regulation, protein synthesis, cell cycle progression, cell proliferation, cell growth and apoptosis in both eukaryotic and prokaryotic cells [1,2]. In the CNS, polyamines are essential for maintaining neurophysiological functions. The regulation of disease processes by polyamines has been described in cerebrovascular diseases (e.g., cerebral ischemia and stroke), neurodegenerative diseases (e.g., Alzheimer’s disease, Parkinson’s disease, and Huntington’s disease), epilepsy and brain tumors [3,4,5,6,7,8]. Moreover, polyamine levels dramatically increase in the brain under pathological conditions, including the traumatic injury, ischemia, neurodegeneration, and excitotoxicity [9,10,11,12,13].

The characteristic symptom of epilepsy is recurrent and unpredictable seizures, which result from abnormal, excessive, and hypersynchronous discharges in a population of neurons. Seizures essentially result from an imbalance between excitement and inhibition of brain function: overexcitement or weakening of inhibition, and the mechanism is complex. Abnormal discharges of neurons during seizures are closely related to ion channels, neurotransmitters, glial cells, heredity, immunity, and inflammation [14,15,16]. As one of the most common serious encephalopathies, epilepsy can induce cognitive impairment, mental health problems, employment restriction, and life inconveniences [17]. Sudden unexpected death in epilepsy (SUDEP) is a major cause of death, and generalized tonic-clonic seizures are the major risk factors for SUPED [18,19]. Although most patients with epilepsy can be effectively treated with drugs, approximately 30% of epileptic cases are refractory to antiepileptic medications. Therefore, more effective treatment options are urgently required [20]. Over the past few decades, the involvement of polyamines in epilepsy has been revealed, but the complex mechanisms remain unclear. In this review, the potential association between polyamines and epilepsy is discussed.

## 2. Polyamines

### 2.1. Biosynthesis of Polyamines

Studies have shown that polyamines are predominantly biosynthesized through the polyamine pathway and the methionine salvage pathway [1,21], and the agmatine pathway was subsequently confirmed in mammals [22]. In mammalian cells, the three polyamines are mainly synthesized from ornithine and methionine in the canonical biosynthesis pathway, which begins with the production of ornithine via conversion from arginine by arginase. Ornithine decarboxylase (ODC) is the first rate-limiting enzyme in polyamine biosynthesis and catalyzes the conversion of ornithine to putrescine. Putrescine is the precursor of spermidine and spermine. An aminopropyl group donated from decarboxylated S-adenosylmethionine (dcSAM) is added to putrescine and spermidine to form spermidine and spermine via spermidine synthase and spermine synthase respectively [23]. As an aminopropyl group donor critical for the synthesis of spermidine and spermine, dcSAM is produced by S-adenosylmethionine decarboxylase (SAMDC) using SAM as a substrate. Methionine adenosyltransferase (MAT) catalyzes methionine to produce SAM in an ATP-dependent process [24,25]. As a potential source of polyamines, agmatine generated from the decarboxylation of arginine by arginine decarboxylase (ADC) can be hydrolyzed by agmatinase (AGM) into putrescine [22,26,27]. In another degradation route, diamine oxidase (DAO) catalyzes the conversion of agmatine to γ-guanidinobutyraldehyde, which is the precursor for the inhibitory neurotransmitter GABA (Figure 2) [28,29].

### 2.2. Catabolism and Interconversion of Polyamines

The higher polyamines, spermine and spermidine, can be converted into lower polyamines through two consecutive enzymatic reactions (Figure 2). The rate-limiting enzyme of polyamine catabolism is spermidine/spermine N^1^-acetyltransferase (SAT1) which acetylates spermidine and spermine into N^1^-acetylspermidine and N^1^-acetylspermine, respectively [30,31]. Then, N^1^-acetylpolyamine oxidase (PAOX) oxidizes acetylated spermidine and acetylated spermine back to spermidine and putrescine, yielding the byproducts 3-acetylaminopropanal (3-AAP) and H_2_O_2_. Another enzyme involved in polyamine catabolism is spermine oxidase (SMOX), which directly converts spermine into spermidine, and produces 3-aminopropanal (3-AP) and H_2_O_2_ during oxidation [32,33]. DAO belongs to the family of copper containing amine oxidases (CuAOs) [34]. In mammalian cells, DAO oxidizes putrescine into 4-aminobutanal (4-AB), which is accompanied by the production of NH_3_ and H_2_O_2_. Then, 4-AB is metabolized to GABA under the action of aldehyde dehydrogenase (ALDH) [35,36]. A putative self-sustaining cell death cycle related to polyamines metabolism has been proposed. Both SAT1 and PAOX catalyze the metabolism of polyamines and produce H_2_O_2_. In addition to causing cytotoxicity, the generation of H_2_O_2_ elevates SAT1 activity and thus positively regulates the process of polyamine metabolism and forms the cycle (Figure 3) [37,38].

### 2.3. Toxic Products of Polyamine Oxidation

Notably, some products of polyamine oxidation are highly toxic to cellular components. These oxidation products, including aldehydes and ROS, are related to cell apoptosis and inhibit the synthesis of DNA and proteins. As mentioned above, the aldehydes generated by oxidases mainly include 3-AP, 3-AAP, and 4-AB. Wood et al. [39] studied and described the toxicities of these aldehydes, first confirming the cytotoxicity of 4-AB and the lack of toxicity of 3-AAP. Both the amino and aldehyde groups of 3-AP determine toxicity, and the amino group confers lysosomotropism. Moreover, 3-AP concentrates in lysosomes and exerts its toxicity by causing lysosomal rupture, which subsequently induces caspase activation and cell death [40,41]. Due to instability, 3-AP and 3-AAP spontaneously convert into acrolein after deamination. As an unsaturated reactive aldehyde, acrolein is predominantly formed from 3-AP produced by SMOX and is less often formed from 3-AAP generated by PAOX (Figure 2) [42].

The toxicity of acrolein has been suggested to be more potent than that of H_2_O_2_ and nearly equal to that of OH [43,44,45]. As a strong electrophile, acrolein shows great reactivity with the sulfhydryl groups of cysteine, histidine and lysine [46]. Acrolein induces protein dysfunction by reacting with lysine residues of proteins to form Nε-(3-formyl-3,4-dehydropiperidino)lysine (FDP-lysine) and Nε-(3-methylpyridium)lysine (MP-lysine) [47,48]. A mitochondrial pathway has been described in a study of acrolein-induced cell death. Acrolein induces depolarization of mitochondrial membrane potential and the liberation of cytochrome c, which cause DNA fragmentation through activation of the caspase cascade reaction [49]. Glutathione (GSH) has been confirmed to have a protective effect on acrolein neurotoxicity [50,51]. Liu et al. [52] found that overproduction of acrolein in a middle cerebral artery occlusion (MCAO) animal model depleted the intracellular GSH and elicited a vicious cycle of oxidative stress, resulting in neurotoxicity through SAT1 activation. As abnormal generation of acrolein usually appears under pathological conditions, it has the potential to be a diagnostic marker in some diseases [53,54].

## 3. Alteration of the Polyamine System in Epilepsy

### 3.1. Alteration of Polyamine Metabolism

The endogenous polyamine system is sensitive to various pathological states, and its homeostasis can be disrupted after brain injuries. In rat brains exposed to different stimuli, the concentration of putrescine increases rapidly, then reduces stepwise and ultimately remains at relatively high levels for days compared to the blank control [13,55,56,57]. The extent of such an increase is positively correlated with the severity of the damage induced. Meanwhile, the levels of spermine and spermidine rapidly decline at first and then recover, even gradually becoming higher during the acute convulsant phase than they were initially in some brain regions [8,13,58]. However, the levels of both spermidine and spermine in the hippocampus are significantly reduced in the chronic phase of pilocarpine-induced epilepsy [59]. Alterations in polyamine levels are mainly affected by changes in the activity and expression of related enzymes [9,60,61]. Therefore, abnormal polyamine synthesis and catabolism play crucial roles in this process.

The increase in putrescine synthesis has been shown to be affected by the upregulation of ODC in response to a variety of brain injuries. In addition to the significantly increased expression and activity of ODC, alterations in other related proteins (such as ODC-AZ, mitochondrial ornithine transporter Ⅱ) have been found to promote the biosynthesis of putrescine in epilepsy models. A marked reduction in the increase in putrescine levels occurred after pretreatment with difluoromethylornithine (DFMO, an irreversible inhibitor of ODC) in epilepsy models, suggesting that upregulation of ODC occupies a significant position in the overproduction of brain putrescine [7,62,63,64,65,66]. However, study results have shown that ODC activity increases markedly within hours of brain injury but falls nearly to control levels within days, while putrescine is maintained at high and relatively stable levels during this period [63]. The transient increase in ODC activity does not seem to be sufficient to entirely explain the long-lasting increase in putrescine levels. In fact, it has been shown that the polyamine interconversion pathway (SAT1/PAOX pathway) is rapidly activated after induced seizures [59,67,68]. Changes in SAT1 show a transient activity elevation and an increase in mRNA expression persisting for days. In these epilepsy models, pretreatment with MDL72527 (an irreversible inhibitor of PAOX) inhibited the SAT1/PAOX pathway and markedly reduced the putrescine content, showing that an enhancement in polyamine interconversion has an important role in the increase in putrescine [59,67,68]. These results are supported by the upregulation of SMOX observed in cerebral ischemic and traumatic brain injury (TBI) models [9,69,70]. As the activity of PAOX is high in nearly all cells and organs in mammals, the interconversion rates seem to not be susceptible to changes in PAOX [71]. In addition, upregulation of SAT1 in some epileptic brain areas is often followed by an increased spermidine/spermine (SPD/SPM) ratio, which is widely used as a metabolic index for polyamine interconversion. Thus, overexpression of SAT1, the rate-limiting enzyme in the SAT1/PAOX pathway, can enhance the catabolism of higher-order polyamines to putrescine. In addition to the involvement of ODC and the SAT1/PAOX pathway in the alteration of polyamine contents during different epileptic periods, studies have also found that a change in SAMDC takes part in this process. SAMDC activity markedly decreases for a short period after ischemia–reperfusion, induced seizures and traumatic brain injury and then gradually recovers and even increases at later stages. The decline in SAMDC activity appears almost coincident with the surge in putrescine [58,61,72,73,74]. In the cortices of patients with refractory epilepsy, Morrison et al. [75] confirmed the significantly increased activity of SAMDC during the chronic phase of epilepsy in humans. Researchers explored the expression of SAMDC by performing real-time RT-PCR and found no significant differences between pilocarpine-treated groups and sham-control groups [59,76]. Overall, changes in SAMDC seem to be mainly manifested by alterations in activity during different epileptic periods. However, there is still a lack of research investigating the activities and expression of enzymes in parallel under the abovementioned pathological conditions.

### 3.2. Heterogeneity of Polyamine Distribution and Changes in Polyamine Homeostasis

Polyamines are abundant in brain tissues, and their distribution and concentrations vary with age and brain regions in both rodents and humans [77,78]. These variations in polyamines maintain the normal function of the brain under physiological conditions. In epileptic brains, heterogeneity of polyamine distributions as well as differences in the extent of polyamine alterations have been reported and are hypothesized to contribute to differential regulations of local brain activities. A relationship between polyamine contents and histological damage has been observed. In kainic acid (KA)-induced epilepsy models, different degrees of increase in putrescine levels were observed in nearly all brain regions. This overproduction of putrescine as a response to KA injection exhibited a positive correlation with the severity of behavioral changes and histological abnormalities and damage [8,13]. In human temporal lobe epilepsy, changes in polyamine metabolism appear to be selective in the hippocampus: spermidine and spermine contents present distinct regional distributions while putrescine contents vary less. The SPD/SPM index increased markedly in the seizure onset areas, and spermine content increased significantly in propagated neocortical areas. Some areas of the hippocampus with increased spermidine levels were characterized by severe gliosis and neuronal loss [7,79]. According to these findings, it is proposed that heterogeneous distributions of polyamines may differentially regulate the local control of excitability through their involvement in cell plasticity and neurotransmitters in epilepsy. A study by Bondy et al. [80] showed a significant increase in ODC activity in brain areas vulnerable to seizure-induced damage, and ODC activity varied more with increasing intensity of electroconvulsive shock within a certain range, partially explaining the regional heterogeneity of polyamine contents in the impaired brain. Regional expression of some enzymes or transporters may also induce such heterogeneity. In healthy rats and epilepsy models induced by pilocarpine intraperitoneal injection, Sadeghi et al. [62] demonstrated differences in the expression of proteins related to putrescine biosynthesis (ODC, ODC-AZ and mitochondrial ornithine transporter Ⅱ) between the hemispheres within groups and between groups. In addition, a sharper increase in the concentration of putrescine has been found to come with the most hyperexcitable site. The elevation of putrescine seems to be related to increased neuronal excitability [56]. Of course, distinct distributions of convulsant agents in epilepsy models may also influence the heterogeneity of polyamine contents in the brain. We speculate that heterogeneities in polyamine alteration may play a role in the generation and development of epilepsy; however, more evidence is needed to confirm this.

## 4. Effects of Polyamines on Epileptic Activity: Inhibiting or Promoting?

### 4.1. The Influence of the Polyamine System on Seizure Susceptibility

Endogenous polyamines are significantly altered and play important roles in the pathogenesis and development of epilepsy. To further explore whether altered polyamine metabolism in response to CNS insults provides neuroprotective effects or is a cause of neurological impairment, transgenic mouse models overexpressing ODC, SAT1 and SMOX have been used for investigation.

Transgenic mice with upregulation of ODC have been found to be protected from both physically and chemically induced seizures and exhibit an elevated seizure threshold. Meanwhile, the mice showed impaired performance in spatial learning and memory tests. These findings are suggested to result from the regulation of N-methyl-D-aspartate receptors (NMDARs) by polyamines [81]. ODC activation is also considered a neuroprotective approach in transgenic cerebral ischemia [82]. Treatment with DFMO induced audiogenic seizures unrelated to changes in whole-brain GABA levels in mice [83]. In electrically kindled rats, DFMO lowered the required stimulation intensity for the first fully kindled state, while putrescine pretreatment at the same injection site showed an inhibitory effect on the development of electrical kindling by regulating synaptic transmission in the brain [55]. In addition to ODC, overexpression of SAT1 also provided neuroprotection against both pentylenetetrazole (PTZ)- and KA-induced seizures, as the neurons showed a damage reduction and elevated seizure threshold in transgenic mice [84,85]. As mentioned previously, the enhanced action of ODC and SSAT contributes to the high accumulation of putrescine in the brain. On the one hand, increased putrescine inhibits NMDARs to exert antiepileptic effects, but on the other hand it impairs learning and memory. Therefore, it is reasonable that related enzymes are involved in changes in susceptibility to epileptogenic stimuli via modulation of the polyamine contents.

SMOX overexpression has also been found to increase susceptibility to seizures [86,87,88,89]. A transgenic mouse model overexpressing SMOX in the cerebral cortex was shown to have a high susceptibility to seizures and damage induced by KA or PTZ. Compared to control mice, SMOX overexpression led to reactive astrocyte activation, which contributed to the release of glutamate (Glu) via the activation of Ca^2+^-permeable α-amino-3-hydroxy-5-methylisoxazole-4-propionic acid receptors (CP-AMPARs) together with increased ROS production, resulting in susceptibility to KA toxicity [89]. As SMOX can oxidize spermine to generate H_2_O_2_, higher ROS production caused by overexpression of SMOX led to an increase in seizure susceptibility in transgenic mice [87]. Pietropaoli et al. [88] further attempted to explain the higher excitotoxic sensitivity. In astrocytes, they found that the increased H_2_O_2_ production induced a reduction in the excitatory amino acid transporter (EAAT) level and upregulation of system x_c_^−^. These changes induced a decrease in EAAT-mediated glutamate uptake and an increase in system x_c_^−^-dependent- glutamate release, leading to substantial glutamate accumulation in the synaptic cleft [88]. Thus, SMOX overexpression can indirectly induce excitotoxic stress and contribute to epileptogenesis by the involvement in excessive glutamate accumulation.

### 4.2. Effects of Polyamines on Epilepsy Progression

Whether the alteration of polyamine levels and the expression and activation of related enzymes after seizures are adaptive neuroprotective responses or the cause of excitotoxicity remains controversial. Both exogenous and endogenous polyamines have been studied in cells and animals to explore their roles in epilepsy pathology. Of course, the effects of polyamines may vary under different experimental conditions.

Elevated levels of cerebral putrescine can be induced by the upregulation of either ODC or SAT1. Increases in ODC activation and putrescine accumulation usually occur after brain stimulation, and some researchers consider them to be a neuroprotective self-regulatory response to stress. Although putrescine failed to exhibit an anticonvulsive effect in previous studies, it slowed the progression of epilepsy. Intra-amygdaloid injection of putrescine before kindling in rats had an inhibitory effect on the development of seizures, whereas pretreatment with DFMO lowered the seizure threshold [55,56]. Grossly elevating putrescine via overexpression of ODC not only significantly enhanced the seizure threshold in response to both chemical and electrical stimuli but also restricted the spread of the elicited seizure [81]. Moreover, cell injury and visible abnormalities were not observed in transgenic mice with high endogenous putrescine levels in the brain [90]. Four hours after the initial PTZ exposure, the application of putrescine delayed the seizure onset time at the second exposure in a Xenopus tadpole PTZ-induced epilepsy model. An atypical pathway mediating the effect was proposed; in this pathway, GABA is converted from putrescine and then activates presynaptic GABA_B_ receptors after PTZ stimulation [91]. Polyamines were suggested to be protective in a mouse model of tuberous sclerosis complex (TSC), as reduced levels of ODC and putrescine worsened neurodevelopmental phenotypes and increased oxidative stress [92].

Exogenous polyamines were demonstrated to be potent convulsants by enhancing CNS excitability in early studies. de Vera et al. [93] found that the intraperitoneal injection of putrescine induced behavioural patterns, including wet dog shakes and motor incoordination, and histological alterations, such as perivascular edema and moderate spongiosis. A positive correlation was identified between the severity of clinical symptoms and the level of putrescine in the brain [93]. In addition to inducing seizures, administration of putrescine into the deep prepiriform cortex also potentiated the convulsant effects of NMDA at a subconvulsant dose [94]. The effects of polyamines vary with the administration dosage. Intraventricular polyamine injections induced sedation and hypothermia at low doses and caused convulsions at higher doses in animals. Spermine showed higher potency and induced extreme hyperexcitation [95]. Exogenous spermine slowed glutamatergic excitatory postsynaptic potential (EPSP) decay, promoted firing on EPSPs and increased membrane rectification during depolarization in thalamic slices of juvenile gerbils. As a result, spermine reduced the seizure threshold and increased the susceptibility to electrical stimulation [96].

## 5. Polyamines and Pathological Changes in Epilepsy

### 5.1. The Role of Polyamines in Regulating Neuronal Excitability

Epilepsy results from an imbalance between the excitability and inhibition of neuronal networks [17]. Epileptic seizures are the direct result of abnormal, excessive and hypersynchronous neuronal discharge. Polyamines are abundant in developing and proliferating tissues and are essential for physiological functions [2]. Polyamine contents are relatively high in the brain tissues of adults. The majority of brain polyamines are stored in synaptic vesicles and astrocytes, allowing for their roles in neuromodulation and neuronal communication.

#### 5.1.1. Polyamines and Ion Channels

Alterations in ion channel expression and function have a large impact on epileptic activity. Many antiepileptic drugs (AEDs) use ion channels as the targets to exert anticonvulsant effects. Disturbances in ion homeostasis are considered to induce excessive neuronal discharges during seizures. Alterations in ion channels during epileptogenesis and epilepsy progression have been well summarized [97,98]. Endogenous polyamines are present at high concentrations in the intracellular compartment and can be released into the extracellular medium. They perform various functions in the brain via the direct regulation of ion channels and receptors, most of which can be regulated by polyamines at nanomolar or micromolar concentrations, with putrescine exhibiting the lowest potency. The complex and diverse effects of polyamines on ion channels and ionotropic glutamate receptors (iGluRs) (Figure 4) have been described in detail [99,100,101] and are therefore not discussed further here.

Intracellular spermine suppressed inward Na^+^ currents and shifted the voltage dependence of voltage-gated Na^+^ channel (VGSC) gating [102]. Fleidervish and colleagues [103] hypothesize that intracellular polyamines are essential for suppressing Na^+^ channel late openings, as DFMO-induced polyamine depletion led to a high magnitude of persistent Na^+^ current (I_NaP_) and generation of spontaneous action potentials in neurons. I_NaP_ plays a significant role in the normal physiological functions of neurons with a slow inactivation property, although its magnitude is less than 1% of the peak transient Na^+^ current (I_NaT_) [104,105]. I_NaP_ participates in action potential bursting, spike threshold determination and modulation of subthreshold and suprathreshold membrane excitability [106,107,108,109,110]. As a result, I_NaP_ amplifies both excitability and inhibition in neurons [111,112]. Basic neurophysiology studies have indicated that abnormally elevated I_NaP_ contributes to neuronal hyperexcitability [113]. Increased I_NaP_ has been found in neurons of different epilepsy models [59,114,115]. The elevation of I_NaP_ after status epilepticus (SE) is thought to be a crucial cause of the subsequent genesis of spontaneous seizures [116,117]. Most antiepileptic drugs acting on VGSCs voltage-dependently inhibit I_NaT_ to suppress neuronal excitability but have no effects on I_NaP_. As an abnormal I_NaP_ has been considered a key pathogenic factor, inhibition of I_NaP_ may be a potential therapeutic target for epilepsy [113,118,119,120]. Du et al. [121] found that reduced I_NaP_ suppressed the seizures caused by mutations affecting axonal K(v)1 channels, confirming an antiepileptic effect of I_NaP_ blockade.

Changes in I_NaP_ rely on regulation of Na^+^ channel subunits at the mRNA or protein levels. Study results have indicated that an upregulation of Na^+^ channel subtypes is a possible explanation for the increase in I_NaP_, while other studies were unable to identify any significant changes in the expression of pore-forming VGSC submits [59,114,115,122]. A polyamine-dependent mechanism has been proposed: the elevation of I_NaP_ in the chronic epilepsy model is attributed to relief from cytoplasmic polyamine blockade, because seizure-induced elevation of I_NaP_ can be completely reversed by saturation with intracellular spermine without significantly modifying Na^+^ channel expression [59,116]. Drug resistance is an important difficulty in the treatment of epilepsy and possible underlying mechanisms have been proposed [20]. Interestingly, recent studies indicate that endogenous polyamines play a key role in resistance generation by regulating I_NaP_. Endogenous polyamines modulate Na^+^ channels in a use-dependent manner. Depletion of intracellular spermine is considered a major factor that induces carbamazepine (CBZ) resistance by increasing I_NaP_ in chronic epilepsy [123]. Blocking I_NaP_ induced anticonvulsant activity and exhibited neuroprotective effects, including prevention of neuron loss and inhibition of hippocampal mossy fiber sprouting [119,120]. These findings suggest that regulation of I_NaP_ by endogenous polyamines is implicated in epileptic activity.

#### 5.1.2. Polyamines and Astrocytic Transmission

Neural networks rely on various neurotransmitter systems. An imbalance between excitatory and inhibitory neurotransmitters, especially the glutamate and GABA systems, can lead to epileptiform activity [124]. Endogenous polyamines are produced in neurons, predominantly accumulated and released by glial cells [125,126,127,128]. Since the functions of astrocytes play key roles in epilepsy, regulation of polyamines may be a mechanism involved in this process.

Loss of GABAergic inhibition is a major mechanism of epileptogenesis. GABA is the major inhibitory neurotransmitter and is mainly synthesized and released from GABAergic neurons. However, it has been found that a considerable amount of GABA can be produced and released from glial cells [129,130]. Endogenous putrescine has been considered the source of astrocytic GABA synthesis [131,132], which is suggested to be the dominant source of gliotransmitter GABA [133]. Thus, metabolism and transport of astrocytic polyamines may influence the levels of GABA and subsequently affect epileptiform activity. Through comparison with the normal group, Laschet et al. [131] found a large amount of putrescine accumulation and a significant acceleration of putrescine catabolism into GABA in astrocytes from epileptic mouse cortices. As a nonneuronal source of GABA in the brain, astrocytes in various regions have been suggested to use diverse amine oxidases to synthesize GABA, including monoamine oxidase (MAO-B) and DAO [129,134,135,136]. Under the action of these enzymes, increased GABA formed by putrescine boosts the Glu/GABA exchange that mediates the release of GABA through astroglial GABA transporters GAT-2/3 and exerts tonic inhibition under epileptic conditions [137]. As the exchange mechanism can be prevented by blocking MAO-B and DAO, putrescine is thought to play a key role in this process [132,137]. It has also been found that glial GABA is released via the bestrophin1 (Best1) channel in the cerebellum and striatum [129]. Notably, both MAO-B and DAO are copper amino oxidases. Copper homeostasis has been explored and confirmed to participate in astrocytic control of neuronal excitability by regulating putrescine catabolism [138].

Gap junction channels (GJCs) mediate direct intracellular communication between astrocytes and/or neurons and are composed of two hemichannels (HCs) or connexons, which consist of six connexins (Cx). Cx43 is expressed abundantly in astrocytes and has been shown to play a predominant role in gap junctional intracellular communication [139]. Research shows that polyamines have a coupling-promoting effect on Cx43 GJCs and that spermine, with four positively charged amino groups, exerts the strongest effect. In astrocytes, cytoplasmic spermine facilitates intracellular communication by preventing Ca^2+^- and H^+^-induced uncoupling of Cx43 GJCs in a concentration-dependent manner [140,141,142]. Therefore, under pathological conditions with H^+^ and Ca^2+^ overload, intracellular spermine has the ability to preserve Cx43-mediated gap junctional communication and maintain astrocyte function. However, Cx43 has been demonstrated to differentially regulate epileptiform activity in different epilepsy models [143]. In a low-Mg^2+^ in vitro epilepsy model, astroglial synchronization mediated by Cx43 gap junctions significantly contributed to the propagation of synchronized neuronal and epileptiform activity [144]. Meanwhile, in the hippocampus of patients and animal models with temporal lobe epilepsy (TLE) with sclerosis, a complete lack of glial gap junction coupling was discovered and was suggested to be a prime cause of TLE [145]. Thus, polyamines may affect epileptiform activity through alterations in astrocytic synchronization and communication by modulating GJCs. This effect may rely on the role of astrocytes in different types of epilepsy. Since polyamines can be interconverted in neurons and are stored in glial cells, it is possible that polyamine conversion influences astrocyte functions.

### 5.2. The Potential Role of Polyamines in Damage and Pathological Changes after Seizures

Epilepsy triggers a cascade of pathological changes, some of which contribute to an enhanced propensity for seizure genesis. Pathological changes in epilepsy refer to neurodegeneration, neuron loss, gliosis, axonal sprouting, acquired channelopathies, microvascular proliferation and others (Figure 5) [146,147]. According to previous studies, endogenous polyamines may be significantly associated with certain pathological processes in epilepsy.

#### 5.2.1. Hypoperfusion/Hypoxia Following Seizures

Seizures can induce severe ischemic/hypoxic attacks [147,148]. A focal decrease in tissue oxygenation occurs following interictal and ictal events [149]. In a study of both animal models and patients with epilepsy, Farrell and colleagues [150] indicated that postictal hypoxia is mediated by vasoconstriction-induced hypoperfusion, which is localized to the brain areas involved in seizures. Cyclooxygenase 2 (COX-2) and L-type Ca^2+^ channels have been confirmed to play important mechanistic roles in vasoconstriction during seizures, and antagonists of the two targets exert inhibitory effects on postictal hypoxia [148,150,151]. A significant increase in the intracellular Ca^2+^ concentration in vascular smooth muscle cells is the root cause of postictal vasoconstriction. According to research, COX-2 is induced during seizure activity and transmits Ca^2+^ signals to downstream L-type Ca^2+^ channels, facilitating inward Ca^2+^ current and mediating enduring pathological vasoconstriction [148,151]. Postictal hypoperfusion/hypoxia has been identified as an important component of seizure-induced brain damage and a cause of various pathological changes, such as neuronal loss, gliosis neuroinflammation, axonal sprouting and blood brain barrier (BBB) dysfunction [147,151]. Therefore, preventing injury from vasoconstriction is necessary for epilepsy treatment.

Polyamines have been suggested to negatively regulate the expression of COX-2 via a post-transcriptional mechanism mediated by eIF-5A, although there is a lack of studies related to the regulation of COX-2 by polyamines in epilepsy [152]. The 3′-untranslated region (3′-UTR) of COX-2 contains three eIF-5A response elements (EREs) that can be bound by the eIF-5A protein. According to study results, DFMO treatment not only induced steady state levels of COX-2 mRNA but also stabilized the COX-2 3′-UTR. This polyamine-dependent suppression of COX-2 RNA seems to be associated with COX-2 3′-UTR destabilization. Thus, polyamines may prevent vasoconstriction via downregulation of COX-2 translation.

Regulation of contractility by polyamines has been found in intestinal and vascular smooth muscle cells [153]. Both intra- and extracellular polyamines have a relaxing effect on contraction via inhibition of inward current through L-type Ca^2+^ channels and a subsequent decrease in intracellular Ca^2+^. Study results have shown that inhibition of spermidine and spermine synthesis enhances Ca^2+^ channel activity [154,155,156]. The effect of polyamines on L-type Ca^2+^ channels is reversible and is determined by the number of positive charges on polyamine molecules. Consequently, spermine is more potent than spermidine, while putrescine has no effect. Moreover, the relaxation appears to occur only in the contraction induced by repetitive action potentials rather than sustained depolarization [157]. These studies demonstrate that polyamines may mediate inhibition of cerebrovascular vasoconstriction by inhibiting L-type Ca^2+^ channels. In addition, polyamines have been found to increase the rapid contraction of smooth muscle through inhibition of myosin phosphatase activity, with the potency correlating with the number of positive charges [158,159]. However, external application of putrescine increased L-type Ca^2+^ currents in mouse neuroblastoma cells [160]. Therefore, the effects of polyamines on L-type Ca^2+^ channels are likely related to cell type.

#### 5.2.2. Gliosis

TLE is the most common form of focal epilepsy in humans and is characterized by hippocampal sclerosis (HS). Reactive gliosis; especially astrocytosis and microgliosis; involves dramatic changes in glial gene expression and cell morphology in epileptic foci. As a typical pathological feature of HS; gliosis has been thought to contribute to the reduction of seizure threshold and the development of epilepsy via a variety of mechanisms; such as induction of neuroinflammation; reduction of K^+^ buffering capacity; impaired gap junction coupling; aberrant neurotransmission and many others [161,162,163]. Indeed; gliosis is present in all forms of epilepsy and has been identified as an integral part of pathological changes [164,165]. Therefore; the regulation of gliosis may have an important influence on epilepsy.

As a histopathological characteristic of epilepsy, astrocytosis has been found to be closely related to polyamine synthesis and metabolism [79,131]. In Dach-SMOX mice, a model overexpressing SMOX in the cerebral cortex, astrocytosis exhibited an increase in cell number and morphological cellular changes consisting of hypertrophy and wide ramification [166]. The effects of chronic activation of neuronal SMOX in the astrocyte process alterations have been revealed, including reduction of spermine levels, increase in Ca^2+^ influx through CP-AMPARs and induction of catalase activity [167]. Meanwhile, enhanced polyamine oxidation induced by SMOX overexpression leads to the production of H_2_O_2_ and acrolein, promoting oxidative stress and neuroinflammation, which are considered key factors in astrocytosis generation. As polyamine accumulation cells, astrocytes respond to neuronal SMOX overexpression and try to regulate polyamine contents, ultimately leading to reactive astrocytosis [166]. Increased synthesis of putrescine by activated ODC was found to induce the expression of glial fibrillary acidic protein (GFAP) in kindling [66]. In a remote astrocytosis model induced by lateral fimbria transection, treatment with DFMO markedly suppressed hippocampal gliosis [168]. Epilepsy is a common phenotypic hallmark of TSC, a quintessential disorder of mechanistic target of rapamycin complex 1 (mTORC1) dysregulation. In Tsc2-RG mice (a transgenic mouse model of TSC), markedly increased ODC activity is attributed to activation of mTORC1 and induces astrogliosis which can be attenuated by DFMO in a dose-dependent manner [169]. However, a decrease in ODC activity and putrescine levels in Tsc2-RG mice is observed to worsen neurodevelopmental phenotypes, indicating protective effects of polyamines in this model [92]. Notably, the promoting effects of ODC on astrocytosis do not hold for each TSC phenotype or lesion condition [169,170].

#### 5.2.3. Neuroinflammation

Evidence is emerging that neuroinflammation may not only be a cause but also a consequence of epilepsy, and complex links between inflammation and seizures have been described [171,172,173]. Increased inflammatory factors have been reported in patients and animal models with epilepsy [174,175,176]. Glial cells, especially astrocytes and microglia, are quickly activated in response to insults to maintain brain homeostasis. However, excessive activation of glia by repeated and long-term seizures can in turn aggravate the severity of seizures [165]. The molecular mechanisms by which inflammation contributes to seizures involve the activation of NMDARs, transcriptional changes related to a low epileptogenic threshold, an increase in glutamatergic neurotransmission and a reduction in astrocyte buffering capacity [173]. Astrocytes, especially microglia, constitute major cellular sources of inflammatory mediators, which play important roles in the crosstalk between the immune and nervous systems in the CNS [177,178,179]. Glial cells can also respond to proinflammatory signals released from immune cells. In this case, mast cells (MCs) have been thought to be key players in these processes [177,178,179].

The effect of MCs in seizures has been explored. A study by Kilinc et al. [180] showed that MCs may exert therapeutic effects in epilepsy because compound 48/80 produced an antiepileptic effect by activating MCs in a dose-dependent manner. Serotonin, one of the mediators released from granules of activated MCs, is suggested to partially or totally mediate the protective effect [180]. Interestingly, the release of serotonin is polyamine-dependent [181]. Antizymes (AZs) are small proteins regulated by polyamines and have inhibitory effects on polyamine synthesis and uptake in mammals [182]. Antizyme inhibitors (AZINs), which are observed to be expressed in MCs, antagonize the action of AZs on polyamines and serve as inducers of ODC activity [181,183]. Kanerva et al. [181] found that activated MCs promoted AZIN2 expression and subsequently contributed to polyamine synthesis. Moreover, polyamine depletion induced by DFMO delayed serotonin release from MCs, indicating that serotonin secretion from MCs can be affected by polyamine synthesis. Meanwhile, potential mechanisms by which polyamines may be involved in the regulation of exocytosis of serotonin-containing MC granules by regulating Ca^2+^ influx or RhoA activity have been proposed [181].

SMOX regulates polyamine catabolism and plays important roles in neuroinflammation. Postischemic neuroinflammation is considered an important mechanism in ischemic brain injury. In rat MCAO models, a significant increase in SMOX expression occurred in neurons and downregulation of SMOX significantly reduced cerebral ischemia injuries. According to the data, the neuroprotective effects of SMOX inhibition are attributed to the inhibition of microglial activation [184]. Alfarhan et al. [185] found that MDL72527 treatment reversed the increase in activated microglial and inflammatory cytokines in excitotoxic retinas, indicating an effect of SMOX inhibition on inflammation reduction. A potential mechanism of neuroinflammation regulation by SMOX can be proposed: activated SMOX induces the generation of acrolein and H_2_O_2_, which have been confirmed to activate microglia, induce proinflammatory signals and elevate oxidative stress [33,186,187].

Oxidative stress plays an important role in seizure-induced damage [188]. ROS levels increase in epilepsy models, and ROS have been suggested to contribute to microglial activation and the production of proinflammatory cytokines. Therefore, scavenging ROS via antioxidant treatment may be a novel approach for neuroinflammation control in epilepsy [189]. Polyamines are considered antioxidants and free radical scavengers [190,191,192,193]. They exert protective effects via direct scavenging of ROS and prevent DNA damage via the induction of DNA conformational changes and blockade of the interaction between DNA and harmful agents [190,193,194,195,196]. Reduced polyamine levels can increase the susceptibility of cells to oxidative stress. From these findings, it can be hypothesized that polyamine synthesis contributes to a reduction in ROS-induced inflammation. ROS production and hypoperfusion/hypoxia conditions in epilepsy also lead to COX-2 activation, which is derived from neurons and results in microglial activation and cytokine production [197,198]. In addition to their inherent antioxidant activity, negative regulation of COX-2 by polyamines has been reported [152]. Thus, inhibition of COX-2 by polyamines may be another pathway to inhibit neuroinflammation.

#### 5.2.4. Synaptic Plasticity

Seizures can induce pathological synaptic plasticity by generating long-lasting changes in synaptic efficacy, which tend to facilitate future epileptogenesis. In the early phases of epileptogenesis, severe pathophysiological alterations appear in the expression of some genes and proteins, GABAergic transmission, the synthesis of neuropeptides, the properties of iGluRs and the availability of some ion channels. Even brief epileptic seizures can generate long-term potentiation (LTP) and increase synaptic efficacy [98,199,200,201,202]. iGluRs are affected by neuronal activity and are involved in regulating synaptic plasticity [203,204,205,206,207]. Continuing neurodegeneration and axonal outgrowth are the prominent features of plasticity changes in the chronic phase of epilepsy, and one typical example is mossy fiber sprouting, which contributes to aberrant excitation of the network [201,202]. A cascade related to alterations in synapsis following epilepsy seizures has been demonstrated. This cascade begins with an increase in the intracellular Ca^2+^ concentration induced by epileptiform activity, and the key steps are the activation of immediate-early genes and genes coding for growth factors, alterations in glutamate receptors and glia, and changes in cytoskeletal proteins [202]. Polyamines may influence the synaptic change related to the cascade through their regulation of neuronal Ca^2+^ channels [160,208]. In addition, a promotive effect of polyamines on axon regeneration has been demonstrated [209,210,211,212,213,214,215]. According to these studies, polyamine biosynthesis is necessary for nerve regeneration, and the possible mechanism involves DNA, RNA, and protein synthesis or posttranslational modifications. Moreover, effects of glial cells on neuroplasticity have been reported due to their roles in neurotransmission and neuroinflammation [216]. As mentioned previously, regulation of polyamines on glial cells may be another pathway affecting neuroplasticity in epilepsy. Even though neuroplastic changes appear to be a mechanism to compensate for seizure-induced damage, they can also facilitate seizure propagation.

#### 5.2.5. Cell Damage and Apoptosis

The mechanism of seizure-induced neuronal death has been identified. Intracellular Ca^2+^ and Na^+^ overload induced by overactivated glutamate receptors leads to swelling and rupture of cells and organelles, activation of proteolytic enzymes and production of free radicals, ultimately leading to neuronal death. Meanwhile, seizures also induce cell death by activating signaling pathways associated with apoptosis [217,218].

Both protective and toxic effects of polyamines on neurons have been demonstrated. Protection by polyamines has been observed in ischemia-induced neuron death. Treatment with polyamines attenuates cell damage and delays neurodegeneration [219,220]. The contribution of ODC activity to ischemia-reperfusion damage has been assessed in rodent models. The results showed that overexpression of ODC delayed the formation and maturation but did not reduce the volume of the infarct induced by permanent occlusion. Increased activity of ODC in the reperfusion phase apparently slowed the development of ischemic damage and reduced infarct volume, while DFMO-treatment and ODC knockdown had the opposite effect, suggesting a neuroprotective role of ODC upregulation in transient cerebral ischemia [82,221,222,223]. Spermine facilitated retention of almost all the activity of the mitochondrial respiratory chain complex and partially protected mitochondria from oxidative stress damage by inhibiting lipid peroxidation. Administration of spermine led to an increase in ecNOS activity and to the preservation of mitochondrial function, which protected against infarctions [219,224]. Within a certain concentration range, Harada et al. [225] found that polyamines prevented cell death in a concentration-dependent manner, with the order of potency being spermine > spermidine > putrescine in cell models of neuronal apoptosis death. Polyamines have also been shown to play a neuroprotective role by suppressing the activation of NMDARs and caspase-3 [225]. Additionally, spermidine protects neurons against neurotoxicity by attenuating oxidative stress, neuroinflammation, and the levels of neurotransmitters [5,6].

Both spermine and spermidine can protect against ROS attack at normal physiological concentrations, but spermine is more effective at lower concentrations. Distinct from GSH in cellular protection against H_2_O_2_, polyamines, as a group of ubiquitous polycations, may prevent oxidative attack by binding to negatively charged molecules (such as DNA, RNA and phospholipids) to form complexes [191,195]. Apart from these findings, endogenous spermine protected antioxidant enzymes in transiently reperfused rat brains, as it reversed the decreases in enzyme activity [226,227].

Polyamines themselves have been confirmed to be cytotoxic. Spermine injection dose-dependently induced neuronal damage in the striatum, and the lesions were reduced by treatment with NMDA and non-NMDA glutamate receptor antagonists [228]. The types of spermine-induced cell death vary with the doses of spermine [229]. Involvement of polyamines in excitatory amino acid-induced neuronal death has also been found [230]. NMDARs antagonists with different mechanisms of action prevented the toxicity of the co-application of spermine and glutamate but not completely, suggesting activation of the non-NMDA receptor pathway [231]. Polyamines also exhibit cytotoxicity by regulating Ca^2+^ influx [232]. In addition, polyamines indirectly mediate toxicity through the toxic products of polyamine oxidation, a process for which amine oxidase is necessary [71,230,231,233,234]. As mentioned previously, acrolein is a toxic aldehyde produced from polyamine oxidation and has been shown to activate microglia and increase oxidative stress in the CNS [187]. The application of antioxidants or amine oxidase inhibitors exerted a protective effect against spermine toxicity in microglia, indicating the involvement of acrolein and H_2_O_2_ in polyamine-induced cell death [235]. Polyamine metabolism is strictly regulated because both a lack and excess of polyamine can be harmful to cells. Polyamine depletion may induce apoptosis via the mitochondrial intrinsic pathway. In polyamine-depleted cells induced by overexpression of SSAT, the application of polyamine analogues belonging to non-SSAT substrates partially restored cell growth and viability, while polyamine oxidase inhibitors showed nonsignificant effects, excluding the effects of ROS and aldehydes and confirming the role of polyamine depletion in apoptosis [236]. In a study by Nitta et al. [237], depletion of polyamines was found to trigger mitochondria-mediated apoptosis and induce not only mitochondrial membrane potential disruption but also activation of caspase-3. In addition, acid-sensing ion channels (ASICs) may be another contributor to ischemic neuronal damage, as extracellular spermine exacerbated ischemic neuronal injury through the sensitization of ASICs to extracellular acidosis [238]. Aberrant regulation of the polyamine system refers to almost all apoptotic pathways, and the complex mechanisms are summarized in Figure 6 [239].

## 6. Conclusions and Future Perspectives

Polyamine homeostasis is essential for maintaining the physiological functions of the CNS. Here, we describe the potential roles of the polyamine system in the regulation of epileptic activity and pathological changes. Although epilepsy is an electrical disorder, it is also a vascular disorder. Blocking postictal hypoxia is necessary for the prevention of brain injury and suppression of epilepsy progression because it plays key roles in neuroinflammation, cell apoptosis and neurodegeneration [240]. As mentioned before, polyamines may be involved in the regulation of vasoconstriction. They also exert protective effects in ischemia and neurotoxic models by scavenging ROS and reducing inflammatory mediators. These results suggest that the use of polyamines may be a preventive strategy to provide neuroprotection against seizure-induced damage. In addition, due to the regulation of ion channels and receptors in the CNS by polyamines, alteration of polyamine levels may affect epileptic activity. Although polyamines themselves, some polyamine analogues and antagonists, and inhibitors of polyamine synthesis and catabolism have been reported to have neuroprotective effects, it is still too soon to identify them as novel therapeutic agents for neuroprotection [68,241,242,243,244,245,246,247]. Notably, both polyamine deficiency and excess are dangerous for cells, suggesting a risk associated with alteration of the polyamine system in treating epilepsy. The roles of polyamines are complex because they involve multiple mechanisms and differ under various brain conditions. An alteration of components of the polyamine system may not necessarily have therapeutic implications in epilepsy, or even elicit unexpected side effects. Nonetheless, human studies have shown a protective effect of spermidine on memory performance and cognitive decline in older adults, and have demonstrated that spermidine is safe and well tolerated in humans [248,249].

Maintenance of endogenous polyamine homeostasis is important for physiological functions in the CNS. Abundant evidence indicates that significant changes occur in the polyamine system in epilepsy. We prefer a condition similar to compensation and decompensation in which proper alteration of the polyamine system may exhibit protective effects against seizure-induced damage, while overregulation may be harmful and contribute to disease progression through various mechanisms. However, what moderate alteration can be made is not clear and still needs to be explored. Further investigations of polyamines in relation to their epilepsy-related pathology, pharmacology, and molecular mechanisms and exploration of analogues/antagonists are also required. Here, we reviewed the potential roles of polyamines in epilepsy, which perhaps provides some new ideas for epilepsy therapy.

## Figures and Tables

**Figure 1 biomolecules-12-01596-f001:**
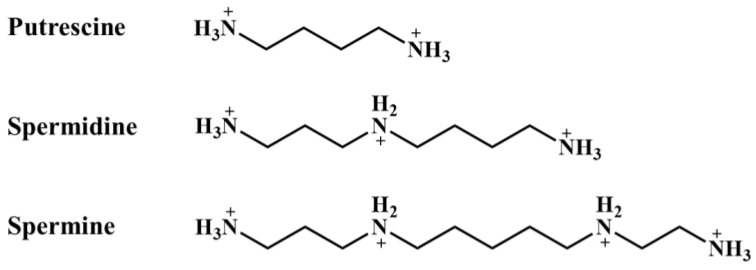
Cationic forms and structures of the three major polyamines in mammalian cells under physiological pH.

**Figure 2 biomolecules-12-01596-f002:**
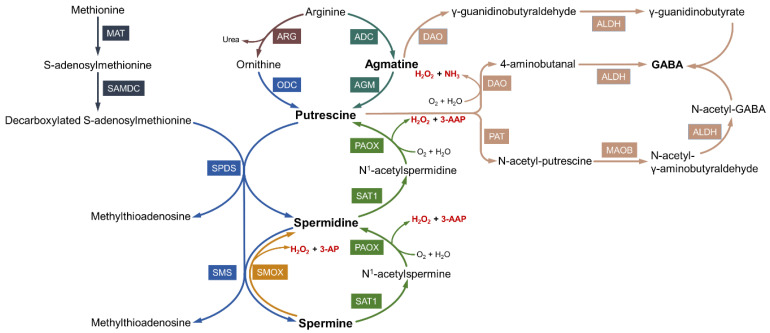
Biosynthesis and metabolism of polyamines.

**Figure 3 biomolecules-12-01596-f003:**
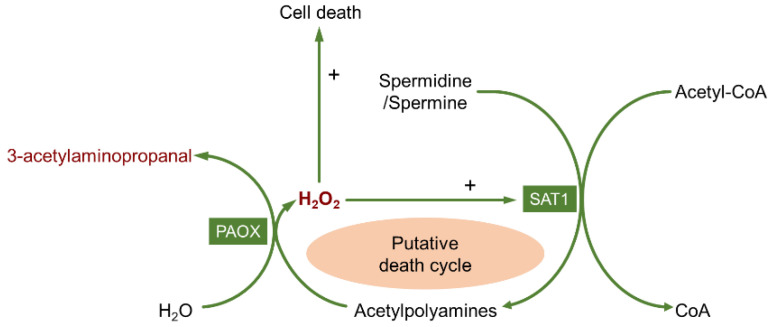
The key role of H_2_O_2_ in the positive cell-death-signal-generating cycle via the action of SAT1 and PAOX. Adapted with permission from ref. [38]. Copyright 2003 Wallace, H.M.

**Figure 4 biomolecules-12-01596-f004:**
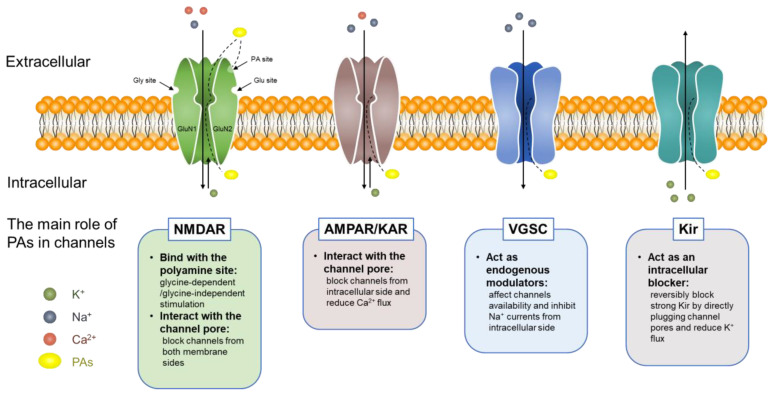
The main role of polyamines in different ion channels and receptors.

**Figure 5 biomolecules-12-01596-f005:**
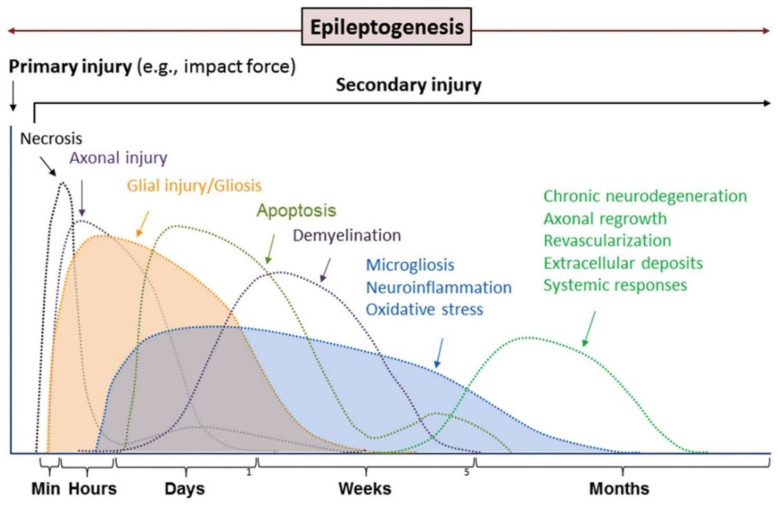
Pathological changes progress and vary with epileptogenesis and the development of epilepsy. Adapted with permission from ref. [146]. Copyright 2019 Elsevier.

**Figure 6 biomolecules-12-01596-f006:**
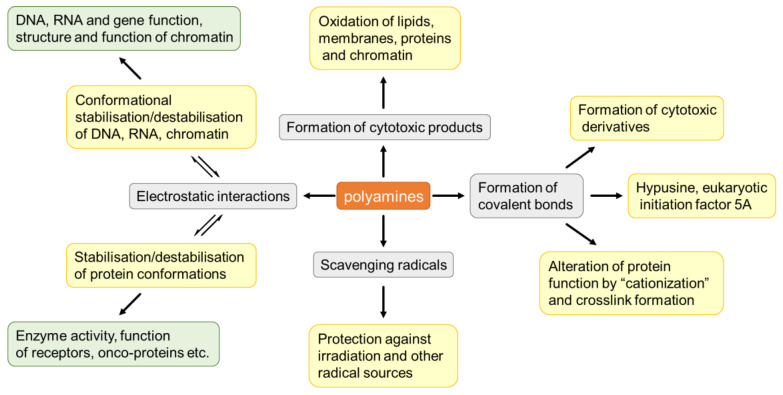
Apoptotic pathways that alteration of endogenous polyamine levels may involve. Adapted from ref. [239].

## Data Availability

Not applicable.

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
