# Peer review of "The Potential Role of Polyamines in Epilepsy and Epilepsy-Related Pathophysiological Changes"

_biomolecules, 2022, doi:10.3390/biom12111596_

Round 1

Reviewer 1 Report

This manuscript is a comprehensive review of the role of polyamines in the development and progression of epilepsy. There has been an increase in interest in the neurological functions of polyamines of late, and this review provides a comprehensive summary of recent findings regarding epilepsy. The review begins with an overview of polyamine metabolism (both the forward and reverse pathways) with an emphasis on the oxidative enzymes involved in the back-conversion pathways, followed by a description of alterations in polyamine metabolism during seizure events. The authors then present a detailed summary of recent research to determine whether polyamines promote or inhibit epilepsy. Figure 3 is a graphical representation of the role of polyamines at 4 important ion channels and receptors, and this figure is followed by presentation of recent data in various neural cell types. There is also a section that deals with inflammation mediated by spermine oxidase, phenomenon that is also relevant in cancer and other diseases. Overall, this article provides a comprehensive overview of research dealing with the role of polyamines in the development and progression of epilepsy, and also suggests that polyamine effector sites may represent promising targets for the targeting of novel therapeutics. The article is very well written and organized, and is thoroughly referenced with recent and pertinent research reports. This article will be extremely useful for anyone with an interest in polyamines in general, and especially for those studying the role of polyamines in neurological disorders.

Author Response

Thank you very much for your time and effort in reviewing the manuscript and giving us an opportunity to revise. We are very grateful for your positive comments and recognition of our work. Changes we have made are highlighted in yellow in the manuscript. In addition to some minor changes that do not influence the content of the paper, we have added a figure (now Figure 3) to make the “putative self-sustaining cell death cycle” mentioned in section 2.2. more intuitive. If there are any other advice for us, we would like to try our best to improve the manuscript.

Thanks again for your comments and suggestions.

Reviewer 2 Report

1) The paper is well written and clear.

2) It looks like a detailed review which may have implications in the future on the study of the pathophysiological mechanisms of epilepsies and on the pharmacological treatments of these diseases.

Author Response

Thank you very much for your time and effort in reviewing our manuscript. We are very grateful for giving us an opportunity to revise. Changes we have made are highlighted in yellow in the manuscript. In addition to some minor changes that do not influence the content of the paper, we have added a figure (Figure 3) to make the “putative self-sustaining cell death cycle” mentioned in section 2.2. more intuitive. If there are any other advice for us, we would like to try our best to improve the manuscript.

Once again, thank you for your kind comments and suggestions.

Reviewer 3 Report

In this review, the authors described the role of polyamines on seizure activity in normal and pathological brain conditions. The review provides a detailed information about physiological effects exerted by polyamines and their role in seizure activity in various brain regions and structure.  This comprehensive review sufficiently describes the problem of polyamines and their influence on seizure activity. This review would bring novel therapeutic suggestions in the field of seizure suppression. Additionally, it clarifies our knowledge on epileptogenesis by presenting a step-by-step mechanisms involved in this process. In my opinion, the paper can be published as it stands.  I have no comments to the paper.

Author Response

We really appreciate your time and effort in reviewing the manuscript and we are grateful for your positive comments and recognition of our work. Changes we have made are highlighted in yellow in the manuscript. In addition to some minor changes that do not influence the content of the paper, we have added a figure (Figure 3) to make the “putative self-sustaining cell death cycle” mentioned in section 2.2. more intuitive. If there are any other advice for us, we would like to try our best to improve the manuscript.

Thanks again for your nice comments.